Psidium guajava L. hydroethanolic extract as endodontic irrigant: phytochemical analysis, antioxidant activity, antimicrobial action and biocompatibility

http://orcid.org/0000-0002-3166-8180 de Carvalho Lara Steffany 1
Meccatti-Domiciano Vanessa Marques 1
da Silva Livia Ramos Dorta 2
Marcucci Maria Cristina 1
Carvalho Cláudio Antonio Talge 2
http://orcid.org/0000-0002-1112-985X Abu Hasna Amjad 2 3 d.d.s.amjad@gmail.com
de Oliveira Luciane Dias 1
1 Department of Biosciences and Oral Diagnosis, Institute of Science and Technology, Campus of São José dos Campos, São Paulo State University , São José dos Campos, São Paulo , Brazil
2 Department of Restorative Dentistry, Endodontics Division, Institute of Science and Technology, Campus of São José dos Campos, São Paulo State University , São José dos Campos, São Paulo , Brazil
3 School of Dentistry, Universidad Espíritu Santo , Samborondón , Ecuador
Moreira Daniel
Electronic publication date: 2025 Apr 14
Publication date: 2025
Volume: 13
Electronic Location ID: e19301
Received 2025 Jan 28; Accepted 2025 Mar 19
Copyright: © 2025 de Carvalho et al.
Copyright year: 2025
Copyright holder: de Carvalho et al.
License: This is an open access article distributed under the terms of the Creative Commons Attribution License, which permits unrestricted use, distribution, reproduction and adaptation in any medium and for any purpose provided that it is properly attributed. For attribution, the original author(s), title, publication source (PeerJ) and either DOI or URL of the article must be cited.
License URL: https://creativecommons.org/licenses/by/4.0/

Keywords: Enterococcus faecalis, Candida albicans, Psidium guajava L., Antimicrobial, Biocompatibility, Endodontics, HPLC

Funding: Coordination for the Improvement of Higher Education Personnel (CAPES) n 88887.829809/2023-00 Institutional Scientific Initiation Scholarship Program–PIBIC National Council for Scientific and Technological Development-CNPQ Edital PROPe 9/2023, n 10195 and CNPQ 313839/2021-2 This work was supported by the Coordination for the Improvement of Higher Education Personnel (CAPES) n: 88887.829809/2023-00; the Institutional Scientific Initiation Scholarship Program–PIBIC of the National Council for Scientific and Technological Development-CNPQ Edital PROPe 9/2023, n 10195; and the Research Productivity Grant of the National Council for Scientific and Technological Development–CNPQ 313839/2021-2. The funders had no role in study design, data collection and analysis, decision to publish, or preparation of the manuscript.

==============================
Background

The search for novel antimicrobial agents in Endodontics is constant to overcoming persistent infections. Psidium guajava L. is a medicinal plant little explored in Endodontics. The aim of this study was to produce hydroethanolic extract of P. guajava L. and to evaluate its phytochemical composition, antimicrobial and antibiofilm action against standard and clinical strains of Enterococcus faecalis and Candida albicans, and cytotoxicity and genotoxicity on human keratinocyte cultures (HaCaT cells). The findings provide new insights into the potential of P. guajava as an alternative endodontic antimicrobial agent, contributing to the development of more effective and biocompatible therapeutic strategies.

Methods

P. guajava hydroethanolic extract was produced using young leaves of guajava and extracted using absolute ethanol and ultrapure water in a ratio (30 g:100 mL). The solid soluble, total flavonoid and total phenols content were determined. The chemical composition was determined via high-performance liquid chromatography (HPLC) analysis, then the free radical suppressive activity was assessed by determining the IC50 value, indicating the concentration required to eliminate 50% of free radicals. Later, the minimum inhibitory concentration (MIC) and minimum microbicidal concentration (MMC) of the extract was evaluated against the strains using clinical and laboratory standards institute (CLSI) guidelines (M27-S4 and M7-A9). Then, the antibiofilm activity was evaluated via MTT (3-[4,5-dimethylthiazol-2-yl]-2,5 diphenyl tetrazolium bromide) assay. Finally, the cytotoxicity of the extract was evaluated via Alamar Blue assay, and the genotoxicity via micronucleus assay on human keratinocyte cultures (HaCaT cells). Data were analyzed using ANOVA and Tukey’s test or Kruskal-Wallis and Dunn’s test.

Results

The soluble solids content in the extract was 3.35%. Using the quercetin standard curve, the total flavonoid concentration was 0.130 ± 0.110 mg/mL. In addition, using standard curve for phenolic acids, the total phenolic concentration was 1.770 ± 1.540 mg/mL. HPLC analysis revealed peaks of rutin, quercetin and kaempferol as major flavonoids in the P. guajava L. extract. The extract demonstrated notable antioxidant activity, with an IC50 value of 10.39 µg/mL. The MMC values ranged 1.04–8.37 mg/mL. The extract at 8.37 mg/mL was effective in reducing the biofilms of standard and clinical strains of E. faecalis and C. albicans after 10 min. Cytotoxicity analysis revealed that all tested concentrations exhibited relatively low toxicity toward HaCaT cells. Genotoxicity assessment via the micronucleus assay indicated minimal DNA damage at all tested concentrations. Overall, P. guajava L. hydroethanolic extract at 8.37 mg/mL is the most effective concentration in reducing the biofilm of the standard and clinical strains of E. faecalis and C. albicans, while maintaining biocompatibility with HaCaT cultures.

Introduction

Root canal infection is classified into primary and secondary/persistent. It is attributed to a variety of microorganisms including Gram-positive, Gram-negative, aerobic and anaerobic bacteria (Narayanan & Vaishnavi, 2010; Endo et al., 2013), with varying degrees of virulence (Machado et al., 2020), in addition to archaea, viruses, and fungi (Siqueira & Rôças, 2022). The success of root canal treatment depends on the effective removal of these microorganisms from the root canal system (Barbosa-Ribeiro et al., 2020) through cleaning and shaping (Schilder, 1974). This is performed using manual or automated endodontic files to exert mechanical action on the infection (Dos Reis et al., 2023; Ragozzini et al., 2024) and endodontic irrigants to exert the chemical action (Abu Hasna et al., 2020b). However, endodontic treatment may fail because of the presence of persistent microorganisms that can resist the disinfection process and reinfect the root canal system (Ng, Mann & Gulabivala, 2011; Karaoğlan, Miçooğulları Kurt & Çalışkan, 2022; Bucchi, Rosen & Taschieri, 2023).

Enterococcus faecalis, as an example of these microorganisms, is a facultative aerobic Gram-positive bacterium present in both primary and persistent/secondary endodontic infections due to its resistance to antimicrobial agents (Santos et al., 2023; Khoury et al., 2024a), ability to adapt to severe environmental changes, and capacity to deeply invade dentinal tubules while tolerating nutrient scarcity (Machado et al., 2020). The endotoxins of E. faecalis, known as lipoteichoic acid (LTA), play a significant role in endodontic infections. These endotoxins contribute to the pathogenesis by triggering inflammatory responses, promoting tissue damage, and sustaining periradicular lesions, thereby complicating the healing process and potentially leading to persistent infections (Oliveira et al., 2022; de Oliveira et al., 2024a).

Also, Candida albicans, is a fungus found in the root canal system during primary and secondary endodontic infections, and it directly associated with the failure of endodontic treatment (Valera et al., 2013; Domingues et al., 2023). C. albicans can firmly adhere to enamel, dentin, and cementum surfaces (Alberti et al., 2021), form biofilms, and invade dentinal tubules because of its thigmotrophic properties, making it resistant to the antimicrobial irrigants used during endodontic treatment (Yoo et al., 2020). It is the most isolated fungus from infected root canals (Mergoni et al., 2018).

Commonly, sodium hypochlorite (NaOCl) is used as endodontic irrigant to combat the endodontic infections (Khoury et al., 2024b), it has a wide antimicrobial and antiendotoxin action (Carvalho et al., 2020; Abu Hasna et al., 2020a), in addition to its ability to dissolve organic tissues (Abu Hasna et al., 2021). However, NaOCl has a high cytotoxicity (Coaguila-Llerena, Raphael da Silva & Faria, 2024) and genotoxicity (Abu Hasna et al., 2022). Conversely, chlorhexidine gluconate (CHX) is another antimicrobial agent used during the root canal treatment (Khoury et al., 2024b), with an effective antimicrobial activity (Mohammadi & Abbott, 2009), however its ability to dissolve organic tissues is limited (Abu Hasna et al., 2021). These limitations of both irrigants make the search for another antimicrobial agents needed.

Phytotherapy has been studied as a means of combating bacteria and fungi directly associated with endodontic infections, showing significant results because of the herbal medicines antimicrobial, antiendotoxin and anti-inflammatory effects (Oliveira et al., 2022; Domingues et al., 2023; Santos et al., 2023; de Lima et al., 2024; Khoury et al., 2024a), besides to its biocompatibility (Ferreira et al., 2021; Dos Santos Liberato et al., 2021; Meccatti et al., 2022; Yu et al., 2022). Psidium guajava L., commonly known as guava, is an American native shrub that thrives in tropical environments worldwide (Gutierrez Montiel et al., 2023). P. guajava L. has numerous applications for treating various conditions, including stomach ailments, diabetes mellitus, cardiovascular diseases, and parasitic infections, highlighting its importance as a subject of study (Tousif et al., 2022; Zhang et al., 2024; de Assis Braga et al., 2025). Furthermore, in endodontics, the use of P. guajava L. was indicated to reduce root canal microflora and root canal failures (Dubey, 2016). In addition, a pilot study evaluated its effectiveness against E. faecalis and C. albicans strains (Baldoni et al., 2023).

It is worth noting that the extraction method, and the used solvent are crucial in determining the efficacy of herbal plants. In the literature, different solvents were used like propylene glycol, ethanol, water, and others (Dos Santos Liberato et al., 2021; Silva et al. 2022; Meccatti et al., 2023; Abubakar & Haque, 2020). Hydroethanolic extraction, which utilizes a mixture of ethanol and water as a solvent, is widely recognized for its efficiency in extracting polyphenols, flavonoids, and other secondary metabolites from plant materials (Plaskova & Mlcek, 2023).

Therefore, the aim of this study was to produce hydroethanolic extract of P. guajava L. and to evaluate its: (I) phytochemical composition; (II) antimicrobial and antibiofilm action against standard and clinical strains of E. faecalis and C. albicans; and (III) cytotoxicity and genotoxicity on human keratinocyte cultures (HaCaT cells). The null hypothesis is that the extract does not exhibit significant antimicrobial and antibiofilm action against the tested microorganisms and has low biocompatibility under the conditions tested.

Materials and Methods

Preparation of the plant extract

The P. guajava L. hydroethanolic extract was prepared using young leaves, manually collected in March 2023 from shrubs in the southern region of São José dos Campos, São Paulo. After collection, the leaves were washed with distilled water, dried in the dark at temperatures between 20–27 °C for 5 days, and stored in a clean, dry environment. The dried plant material was ground using a blender. The solvent used for extraction was absolute ethanol (ethyl alcohol 99.5%; Merck, Darmstadt, Germany) and ultrapure water obtained from a Milli-Q® system (EtOH:H2O/50:50). The ratio was 30 g of plant material per 100 mL of solvent, with an extraction period of 48 h. The extracts were filtered in two stages: first, using a common article filter with micro-pores, the same used to filtrate coffee, made from bleached or unbleached cellulose fibers with pore size generally ranges from 15 to 25 µm, to remove solid residues and then sterilized using a 0.22 µm membrane filter (MilliporeSigma, Millex®) (Baldoni et al., 2023). The extract was stored for further analysis.

Soluble solid content of P. guajava L. hydroethanolic extract

Three empty 25 mL beakers were weighed, and their weights were recorded. Then, 5 mL of the extract were pipetted into each beaker (in triplicate) and dried at 80 °C for 24 h. After drying, the beakers were cooled in a desiccator and weighed again. The soluble solid content of the extracts was quantified using the formula (Meccatti et al., 2023; de Oliveira et al., 2024b):

%solublesolids(m/V)=(m−b)×100/Va

%solublesolids(m/m)=%solublesolids(m/V)/density

where: b = beaker mass; m = final mass of the extract after drying; Extract density = m/V (mass of the 5 mL aliquot weighed, and V is the volume of 5 mL).

The concentration of the extract was determined using the weight/volume (w/v) method. First, the soluble solid content was measured, and the corresponding values were converted from % v/v to w/v. This conversion was achieved by multiplying the percentage by 10, considering that 1% v/v is equivalent to 10 mg/mL. Consequently, the final concentrations are expressed in w/v in the results.

Total flavonoid content determination of P. guajava L. hydroethanolic extract

To determine total flavonoid content, a stock solution was prepared using 100 µL extract in a 10 mL volumetric flask filled with 9,900 µL of methanol (absolute methyl alcohol; Êxodo Científica, Sumaré, Brazil) to the meniscus in a proportion of (1:99). The procedure was performed in triplicate. A 200 µL aliquot from the stock solution was transferred to a 10 mL flask containing ~5 mL of methanol, followed by 200 µL of aluminum chloride (AlCl3), and the volume was completed with methanol in a proportion of (2:2:98). The resulting solution was agitated and incubated in a water bath (Generalmed, São Paulo, Brazil) for 30 min at 20 °C. After adjusting the meniscus, absorbance was read using an ultraviolet-visible (UV-Vis) spectrophotometer at 425 nm (Micronal B-582; Micronal São Paulo, Brazil). Total flavonoid concentration (in mg/mL), expressed in quercetin, was calculated using a calibration curve (Cristina Marcucci et al., 2021; Meccatti et al., 2023; de Oliveira et al., 2024b).

Total phenol content determination of P. guajava L. hydroethanolic extract

In a volumetric flask of 100 mL, 1 mL of the extract was transferred and diluted in a 1 mL of ethanol (99.5% ethyl alcohol-Merck Darmstadt, Germany), then diluted to volume with distilled water (obtained using the Milli-Q® system) while stirring (stock solution) (10 μg/mL). From this point, the procedure was conducted in triplicate. A 0.2 mL (200 μL) aliquot was transferred to a 10 mL volumetric flask (1:50) containing 5 mL of distilled water, with the addition of 800 μL of Folin–Ciocalteu reagent (Sigma-Aldrich, St. Louis, MO, USA). The mixture was stirred for a few seconds, and between 1 and 8 min later, 1.2 mL of 20% sodium carbonate-tartrate buffer solution was added. The volume was completed with distilled water to the meniscus, and the solution was maintained in a water bath (Generalmed, São Paulo, Brazil) at 20 °C. After 2 h, the final volume was adjusted by adding distilled water up to the 10 mL mark at 20 °C with agitation for a few seconds, and the absorbance was measured at 760 nm using a UV-Vis spectrophotometer (Micronal B-582). The total phenol content was determined using the calibration curve equation. A stock solution of phenol (standard) was used to prepare the calibration curve for subsequent quantification (Cristina Marcucci et al., 2021; Meccatti et al., 2023; de Oliveira et al., 2024b).

High-performance liquid chromatography analysis of P. guajava L. hydroethanolic extract

High-performance liquid chromatography (HPLC) was used to characterize the marker content and phytochemical profile of the extracts. The extract used in this analysis is not diluted, it is the same extract prepared inutility from the guava leaves. The analysis was performed on an HPLC system with a diode-array detector (HPLC-DAD) and an automatic injector (D-7000; Merck-Hitachi, Darmstadt, Germany). Chromatographic conditions included a mobile phase consisting of aqueous formic acid solution (95:5, solvent A) and chromatographic-grade methanol (Merck, Darmstadt, Germany, solvent B). The flow rate was set to 1 mL/min with a linear gradient starting at 0% B and ending at 70% B over 50 min. Detection wavelengths of 280 and 340 nm were used (Meccatti et al., 2023; de Oliveira et al., 2024b).

Determination of free radical suppressive activity of P. guajava L. hydroethanolic extract (Antioxidant Activity)

Samples were prepared at different dilutions based on previously determined soluble solid contents. The required extract volume was calculated to obtain 1% V/V and 0.01% V/V dilutions. For 1% V/V dilution, the appropriate aliquot of extract solution was added to a 10 mL flask, and the volume was adjusted with ethanol. The 0.01% V/V dilution was prepared using a serial dilution method. Eleven test tubes were labeled from 0 to 10, and the ethanol and extract solutions were added in sequence using 0, 40, 80, 120, 160, 200, 240, 280, 320, 360, 400 µL of the extract, and adjusted with ethanol to obtain 1,000 µL. In addition, tube 0 served as the control, containing only 100% DPPH (1,1-diphenyl-2-picrylhydrazyl) without plant extract. Also, 1,000 µL of DPPH was added to tube 1, and the reaction time was recorded for 1 min before sequentially adding DPPH with the same volume (1,000 µL) was added to the remaining tubes at 1-min intervals, with periodic shaking. After 30 min, spectrophotometric readings (Micronal B-582) at 517 nm were taken. Each measurement was performed in triplicate, and absorbance values were plotted as Absorbance (%) vs. extract concentration (µg/mL). The IC50 value indicating the concentration required to eliminate 50% of free radicals, was determined using the least squares method (Alves et al., 2010; Veiga et al., 2017).

Determination of minimum inhibitory concentration (MIC) and minimum microbicidal concentration (MMC) of P. guajava L. hydroethanolic extract

One standard (ATCC 4083) and two clinical strains (denominated 2 and 4) of E. faecalis and one standard (ATCC 18804) and two clinical strains (denominated 14 and 60) of C. albicans were used in this study. E. faecalis was cultured (37 °C/24 h) on Brain Heart Infusion agar (BHI ), and C. albicans strains were cultured (37 °C/24 h) on Sabouraud-dextrose agar (SD-Kasvi, São José dos Pinhais, Brazil). Microbial suspensions were prepared by diluting colonies of the respective strains in sterile saline solution (0.9% NaCl) and homogenized in a vortex mixer for 10 s to standardize the microbial solution according to each protocol.

The broth microdilution method, according to the Clinical and Laboratory Standards Institute (CLSI), standards M27-S4 and M7-A9, was used to determine the MIC and MMC of the extract for each microbial strain. The inoculum was standardized in a spectrophotometer (Micronal B-582). For E. faecalis, the wavelength of 760 nm with an optical density of 0.298 was used, resulting in a standard suspension of 1 × 106 bacterial cells/mL. For C. albicans, the wavelength of 530 nm with an optical density of 0.284 was used, resulting in a standard suspension of 1 × 106 yeast cells/mL.

In separate microplates (Kasvi K12-096; Kasvi, São José dos Pinhais, Brazil), a total of 10 serial dilutions (1:2) of the extract were performed in culture media: Mueller Hinton broth (HiMedia®, Mumbai, India) for E. faecalis, and RPMI 1,640 broth (with glutamine, without bicarbonate, and with phenol red indicator) (INLAB) for C. albicans. Aliquots of 100 µL of each microbial suspension were added to all wells. After 24 h incubation at 37 °C, the MIC was determined as the last well without turbidity indicating microbial growth.

To determine MMC, aliquots from all wells were plated on BHI agar and incubated at 37 °C for 48 h. The MMC was identified as having the lowest concentration without colony growth. A vehicle control group (EtOH:H2O/50:50) was included to evaluate its potential interference with the extract’s antibacterial activity.

Antibiofilm activity of P. guajava L. hydroethanolic extract via MTT analysis

For E. faecalis, in 96-well microplates, a volume of 100 μL/well of BHI broth (Kasvi) was added. After preparation and standardization with a spectrophotometer (107 cells/mL), the bacterial suspension was added to the microplates (100 μL/well), already containing culture medium, for a total of 200 μL/well. For C. albicans, the standardization was performed using a spectrophotometer to obtain 107 cells/mL. Subsequently, 200 μL/well of the adjusted C. albicans suspension was added to the microplates and incubated at 37 °C for 90 min to allow initial cell adhesion to the wells. Afterward, the supernatant was discarded, and BHI broth was added. The plates were incubated at 37 °C for 48 h to allow biofilm formation, with the medium being replaced after 24 h (Marques Meccatti et al., 2022; Santos et al., 2023; de Lima et al., 2024).

After biofilm formation, they were exposed for 10 min to different concentrations of the plant extract (1.04. 2.09, 4.18 and 8.37 mg/mL), which were determined based on the results of the MIC and MMC tests. Culture medium was used as a negative control, while 2% CHX (Biofórmula Manipulação, São José dos Campos, SP, Brazil) and 2.5% NaOCl (Asfer, São Caetano do Sul, São Paulo, Brazil) were used as positive controls for all biofilm analyses. Each experimental group consisted of n = 10. Later, saline solution was added and discarded to wash the wells and remove non-adherent cells affected by the treatments. The microbial cell viability test was performed by adding 100 μL of MTT solution (3-(4,5-Dimethylthiazol-2-yl)-2,5-Diphenyltetrazolium Bromide) (Sigma-Aldrich) to each well. The plate was incubated in the dark at 37 °C for 1 h. After incubation, the MTT solution was removed, followed by the addition of 100 μL of dimethylsulfoxide (DMSO). The plate was incubated again at 37 °C for 10 min and then placed on a shaker under constant agitation for another 10 min. Finally, optical densities (OD) were measured using a microplate reader at 570 nm, and the obtained OD values were converted into percentages of metabolic activity.

Cytotoxicity evaluation of P. guajava L. hydroethanolic extract

The cytotoxicity analysis of the P. guajava L. extract was performed on human keratinocyte cell lines (HaCaT) obtained from the Rio de Janeiro Cell Bank–Associação Técnico Científica Paul Ehrlich (APABCAM–RJ). The cell lines were cultured in Dulbecco’s Modified Eagle Medium (DMEM–GC Biotecnologia, Cotia, Brazil) with high glucose concentration, supplemented with 10% fetal bovine serum (FBS) (Invitrogen, Waltham, MA, USA), and maintained in cell culture flasks (Kasvi, Brazil) at 37 °C in a humidified incubator with 5% CO2. Once sufficient cell quantities were reached, the cells were subjected to the respective tests.

The cell monolayer was detached from the culture flask using trypsin. The cultures were transferred to 96-well microplates (Kasvi) at a concentration of 2 × 104 viable cells per well and cultured in 200 µL of DMEM + 10% FBS. The plates were incubated at 37 °C with 5% CO2 for 24 h to allow cell adhesion. Following the incubation period, the cells were exposed for 10 min to different concentrations of the extract (1.04. 2.09, 4.18 and 8.37 mg/mL) showing antimicrobial activity. The negative control group contained only DMEM medium in its liquid 1× form, while the positive control groups included 2% CHX and 2.5% NaOCl.

The cells were subjected to the Alamar Blue assay, using a stock solution of 5 g of Resazurin sodium salt (Sigma-Aldrich, Jurubatuba, Brazil) dissolved in 500 mL of PBS. The solution was added to the 96-well microplates at a volume of 100 µL/well, followed by light-protected incubation at 37 °C with 5% CO2 for 4 h. The solution was then discarded, and 100 µL/well of dimethyl sulfoxide (DMSO–Sigma) was added. After 10 min of incubation and agitation on a shaker, the absorbance of the wells was measured with a microplate reader at a wavelength of 570 nm. The optical density (OD) values obtained were converted into percentages of cell viability.

Cytotoxicity was evaluated according to ISO 10993-5:2009 guidelines for in vitro cytotoxicity assessment, which classifies materials based on their impact on cell viability. A reduction in viability below 70% indicates a cytotoxic effect (ISO 10993-5, 2009).

Genotoxicity evaluation of P. guajava L. hydroethanolic extract

The micronucleus test was conducted in accordance with the OECD 2016 guideline (TG 487) for the HaCaT human keratinocyte cell line. For the test, 5 × 105 cells/mL were cultured in 24-well plates for 24 h at 37 °C in a 5% CO2 atmosphere. After this period, the cells were exposed to the extract at different concentrations (1.04. 2.09, 4.18 and 8.37 mg/mL), 2% CHX, and 2.5% NaOCl for 24 ho. Subsequently, the cells were incubated with cytochalasin B (Sigma-Aldrich) at a concentration of 6 µg/mL for 24 h at 37 °C in a 5% CO2 atmosphere to inhibit cytokinesis and induce the accumulation of binucleated cells. The cells were subjected to hypotonic shock and fixed in a methanol and acetic acid solution (3:1) for 10 min, a procedure repeated three times. The wells were stained with the addition of one drop of DAPI for 5 min. The ethyl methanesulfonate (EMS) that induces the formation of micronuclei was used as a control group.

Micronuclei were analyzed using fluorescence microscopy (Leica Microsystems, Wetzlar, Germany) at 400× magnification, evaluating 2,000 cells per well in at least two independent experiments. Micronuclei were identified as DNA-containing structures in the cytoplasm, clearly separated from the main nucleus, surrounded by a nuclear membrane, and occupying an area smaller than one-third of the main nucleus. Cells with fewer than five micronuclei were counted (Abu Hasna et al., 2022).

Statistical analysis

Data normality was assessed with the Shapiro-Wilk test. Normally distributed data were analyzed using ANOVA and Tukey’s test; non-parametric data were analyzed using Kruskal-Wallis and Dunn’s test. GraphPad Prism 5.0 software was used, with a significance level of 5%.

Results

Soluble solid, total flavonoid total phenol content determination of P. guajava L. hydroethanolic extract

The soluble solids content in Psidium guajava L. hydroethanolic extract was 3.35%. Using the quercetin standard curve, the total flavonoid concentration was 0.130 mg/mL (0.013%), with a standard deviation of 0.110 mg/mL (0.011%). In addition, using standard curve for phenolic acids, the total phenolic concentration was 1.770 mg/mL (0.177%), with a standard deviation of 1.540 mg/mL (0.154%).

High-performance liquid chromatography analysis of P. guajava L. hydroethanolic extract

The chromatographic analysis by HPLC revealed peaks of rutin at retention time of 10.33 min, quercetin at retention time of ~17.83 min, and kaempferol at retention time of ~24.40 min in the extract of P. guajava L., as shown in Fig. 1.

Figure 1 The chromatographic analysis by HPLC of P. guajava L. hydroethanolic extract.

Determination of free radical suppressive activity of P. guajava L. hydroethanolic extract (antioxidant activity)

The concentration required to eliminate 50% of free radicals (IC50) was 10.39 µg/mL for the P. guajava L. extract, with a standard deviation of 2.90 µg/mL.

Determination of minimum inhibitory concentration (MIC) and minimum microbicidal concentration (MMC) of P. guajava L. hydroethanolic extract

The extract demonstrated bactericidal and fungicidal activity against all the tested strains of E. faecalis and C. albicans. The MMC values ranged 1.04–8.37 mg/mL as shown in Table 1, respectively. It was not possible to identify the minimum inhibitory concentration (MIC) values due to turbidity caused by the color of the extract, which hindered visual reading. Therefore, in this study, the MBC values were considered for biofilm evaluation.

Table 1 MMC value of the extract against strains of Enterococcus faecalis and Candida albicans.

Strains	MMC (mg/mL)	
E. faecalis ATCC	1.04	
E. faecalis 2	2.09	
E. faecalis 4	1.04	
C. albicans ATCC	8.37	
C. albicans 14	2.09	
C. albicans 60	2.09	

Antibiofilm activity of P. guajava L. hydroethanolic extract via MTT analysis

All the tested concentrations were effective against E. faecalis ATCC after 10 min except of 1.04 mg/mL which had no statistically significant difference (P < 0.0001) with the control group. However, the extract at 8.37 and 2.09 mg/mL reduced the biofilm formation by 51.62% and 46.58%, respectively, being as effective as NaOCl and CHX without a statistically significant difference (P < 0.0001). Similarly, all the tested concentrations were effective against E. faecalis clinical strain 2 after 10 min except of 1.04 mg/mL which had no statistically significant difference (P < 0.0001) with the control group. However, the extract at 8.37, 4.18 and 2.09 mg/mL reduced the biofilm formation by 53.44, 44.41 and 33.74 %, respectively, with a statistically significant difference with the control group (P < 0.0001). Even more, all the tested concentrations were effective against E. faecalis clinical strain 4 after 10 min. The extract at 8.37 and 2.09 mg/mL reduced the biofilm formation by 69.27, 61.24%, respectively, being as effective as NaOCl and CHX without a statistically significant difference (P < 0.0001) as shown in Fig. 2.

Figure 2 The microbial load after treatment with the groups.

The letters (A, B, and C) denote statistically significant differences among the experimental groups.

The extract at 8.37 mg/mL was effective against C. albicans ATCC strain after 10 min, in which it reduced the biofilm formation by 60.90%, being as effective as NaOCl and CHX without a statistically significant difference (P < 0.0001). Furthermore, the extract at 8.37 and 4.18 mg/mL was effective against C. albicans clinical strain 14 after 10 min, in which it reduced the biofilm formation by 63.27 and 55.39%, being as effective as NaOCl and CHX without a statistically significant difference (P < 0.0001). In addition, the extract at 8.37 and 4.18 mg/mL was effective against C. albicans clinical strain 60 after 10 min, in which it reduced the biofilm formation by 50.14 and 38.68% without a statistically significant difference (P < 0.0001) as shown in Fig. 2.

Cytotoxicity evaluation of P. guajava L. hydroethanolic extract

All the tested concentrations of the extract were less cytotoxic than NaOCl and CHX with a statistically significant difference (P < 0.0001). However, they had a statistically significant difference (P < 0.0001) as well with the control group (Fig. 3).

Figure 3 The cellular viability after treatment with the groups.

The letters (A, B, and C) denote statistically significant differences among the experimental groups.

Genotoxicity evaluation of P. guajava L. hydroethanolic extract

All the tested concentrations of the extract presented a relatively low quantity of micronuclei in comparison to the EMS (the micronuclei former), with a statistically significant difference (P < 0.0001), still all the tested concentrations of the extract were less genotoxic than NaOCl and CHX with a statistically significant difference (P < 0.0001) as shown in Fig. 4.

Figure 4 The micronuclei load after treatment with the groups.

The letters (A, B, and C) denote statistically significant differences among the experimental groups.

Discussion

The search for effective alternative antimicrobial agents against microorganisms associated with persistent endodontic infections is constant. This study aimed to produce hydroethanolic extract of P. guajava L. and to evaluate its phytochemical composition, antimicrobial and antibiofilm action against standard and clinical strains of E. faecalis and C. albicans, and cytotoxicity and genotoxicity on human keratinocyte cultures (HaCaT cells). It was found that in some concentrations, the extract was effective against standard and clinical strains of the tested microorganisms and was biocompatible on the tested cell culture, for that the null hypothesis was partially rejected.

The soluble solids content in Psidium guajava L. hydroethanolic extract was 3.35%, serving as the basis for determining flavonoid and phenolic concentrations. The flavonoids content in the extract used in this study was lower than reported in other studies in the literature. Despite this, it still exhibited notable antimicrobial activity. It is well established that all flavonoids are phenols, but not all phenols are flavonoids; which explains why the phenol content always exceed the flavonoid content (Chaves et al., 2020). In comparison, one study reported a flavonoid content of 1.91 mg/mL, while the present study obtained a result of 0.130 mg/mL. Similarly, the total phenol content also differed, with values diverging from 309.91 mg/mL in the literature (Paiva et al., 2023) to 1.170 mg/mL in the present study. According to the findings of a recent study, the presence of flavonoids such as quercetin is influenced by environmental factors like altitude, temperature, humidity, soil, and pH which influence flavonoid biosynthesis. A recent study established a correlation indicating that higher altitudes are associated with higher flavonoid content (Majhi et al., 2023). However, despite the reduced flavonoid levels, the extract demonstrated antimicrobial activity, reinforcing the potential role of other bioactive compounds in its activity.

Among these compounds, quercetin was identified in this study, is among the main components of the P. guajava L. extract. The identification of this compound at a retention time of 17.83 min aligns with the findings of another study, in which the same compound was identified at a retention time of 22.14 min (Díaz-de-Cerio et al., 2016). The identification of phytochemical compounds in this study, performed using high-performance liquid chromatography, identified three different compounds: rutin, quercetin, and kaempferol, which are known for their strong antioxidant and antimicrobial properties. These flavonoids are known for their ability to scavenge free radicals, thereby reducing oxidative stress and preventing cellular damage. Quercetin exhibits antibacterial activity by disrupting bacterial cell membranes, inhibiting essential enzymes, and interfering with quorum sensing mechanisms that regulate biofilm formation (Nguyen & Bhattacharya, 2022). Rutin is recognized for its strong antioxidant properties, beside to its antimicrobial action which is attributed to different mechanisms (Ivanov et al., 2022; Muvhulawa et al., 2022), while kaempferol has been associated with modulation of immune responses and inhibition of microbial growth (Periferakis et al., 2022; Guan et al., 2024).

However, there are reports in the literature identifies different other compounds, such as vescalagin (RT: 7.71 min), catechin (RT: 9.58 min), ellagic acid (RT: 16.26 min), naringenin (RT: 26.72 min), quercetin glucoside (RT: 34.78 min), reynoutrin (RT: 37.41 min), and chrysoeriol (RT: 86.90 min) in guava leaves (Díaz-de-Cerio et al., 2016; Bezerra et al., 2018; Gutierrez Montiel et al., 2023). These phenolic compounds enhance the antimicrobial and antioxidant potential of the extract. They work by altering bacterial membrane permeability, inhibiting essential metabolic pathways, and preventing microbial adhesion and biofilm development. Their antioxidant capacity helps to neutralize reactive oxygen species (ROS), reducing inflammation and promoting tissue healing.

In endodontics, where persistent infections and biofilm-associated resistance pose significant challenges, the presence of these flavonoids and phenols in P. guajava L. extract suggests a promising alternative to conventional antimicrobial agents. By leveraging the synergistic action of these bioactive molecules, the extract shows potential as a natural, biocompatible antimicrobial agent that could improve root canal disinfection and treatment outcomes.

The antioxidant potential of the extract is directly related to the ability of its compounds to neutralize free radicals and the method used for extraction (Selestino Neta et al., 2017). The antioxidant activity results of the P. guajava extract in this study align with those of (de Souza et al., 2021), who, despite using essential oil from P. guajava, reported an IC50 of 8.94 µg/mL, a value very similar to the 10.39 µg/mL obtained in this study, even with different extraction methods. Another study also evaluated the antioxidant activity of P. guajava L. extracts obtained using methanol and hexane, showing that results varied depending on the extraction solvent, with IC50 values of 10.33 and 16.72 µg/mL, respectively (Purba & Paengkoum, 2022).

The literature indicates that hydroethanolic extracts are more efficient as extraction agents, as the quantity of compounds extracted with this method was higher compared to other solvents (Morais-Braga et al., 2017). Furthermore, the antioxidant capacity of the extract is also related to the ethanol concentration used in the extraction of guava leaves. In a study testing ethanol concentration of 30%, 50%, and 70%, it was observed that higher ethanol concentrations resulted in greater antioxidant activity. The IC50 for the extract obtained with 70% ethanol was 1.40 µg/mL, while for the 30% ethanol extract, it was 2.70 µg/mL (Park et al., 2024).

In the present study, the P. guajava L. extract demonstrated antimicrobial and antibiofilm properties against E. faecalis, consistent with the findings of Dubey (2016). Although Dubey used the inhibition halo test, they also observed antimicrobial activity against this strain and compared it to 2.5% NaOCl. Another study highlighted the antimicrobial properties of ethanolic extract of P. guajava L. extract against an ATCC strain of E. faecalis, but it required a concentration of 20 mg/mL of the extract to inhibit bacterial cell proliferation (Elchaghaby, Abd El-Kader & Aly, 2022). This concentration is significantly higher than the minimum microbicidal concentration found in the present study, which ranges from 1 to 2 mg/mL.

Furthermore, in another study, it was found that P. guajava L. aqueous extract at 20% and 30% concentrations was effective in reducing the colony forming units of Streptococcus mutans, Lactobacillus acidophilus, and E. faecalis after 5 min and 3 h (Vignesh et al., 2017). In addition, a pilot study evaluated its effectiveness against E. faecalis strains and found that it showed microbicidal potential against strains of E. faecalis, being MIC of 0.20% (Baldoni et al., 2023). In the present study, the hydroethanolic extract at 8.37 mg/mL was effective in reducing the biofilm of E. faecalis ATCC and two clinical strains after 10 min of contact between 51.62% and 69.27% being as effective as NaOCl and CHX.

The antifungal potential of P. guajava L. extract has also been explored in the literature. This antifungal activity is attributed to the presence of tannins, as demonstrated by Mailoa et al. (2014), who confirmed the antimicrobial power of the extract against C. albicans and other microorganisms such as Escherichia coli, Pseudomonas aeruginosa, Staphylococcus aureus, and Aspergillus niger. The concentration of tannins was positively correlated with the antimicrobial efficacy of the extract. Moreover, various solvents, including hexane, ethyl acetate, ethanol, and methanol, have been used to obtain P. guajava L. extracts, all showing effective action against different bacterial strains and C. albicans (Jebashree et al., 2011). These findings align with the results of the present study, which demonstrated antibiofilm activity against both standard (ATCC) and clinical strains of C. albicans at lower concentrations compared to the studies mentioned, with an exposure time of 10 min. However, the present study’s results contrast with those of Baldoni et al. (2023), who reported no antifungal activity against ATCC or clinical strains of C. albicans. In the present study, a concentration of 8.3 mg/mL was effective against the ATCC strain of C. albicans, while a concentration of 2.0 mg/mL was sufficient to inhibit the growth of the clinical strain of C. albicans.

In the present study, HaCaT cells were used to ebalute the biocompatibility of the P. guajava L. hydroethanolic extract, as HaCaT are immortalized human keratinocytes that closely resemble normal epithelial cells and play a crucial role in the healing and immune response of the oral mucosa and periapical region (Gursoy et al., 2016; Colombo et al., 2017). Their use as a well-established model for evaluating biocompatibility, cytotoxicity, and genotoxicity allows for a comprehensive assessment of the extract’s potential impact on periapical health (Yu et al., 2022; Meccatti et al., 2023; Miranda et al., 2024).

When evaluating the cytotoxicity of P. guajava L. extract on human keratinocyte cell cultures, all the tested concentrations of the extract in the present study were less cytotoxic than NaOCl and CHX with a statistically significant difference (P < 0.0001). However, they had a statistically significant difference (P < 0.0001) as well with the control group. These fidnings are aligned with those of another study (Alves et al., 2023), that reported a 20% reduction in cell viability using a concentration of 1.25 μg/mL of P. guajava L. extract, tested with a 24-h contact time.

To date, no articles have been found in the consulted literature evaluating the genotoxic effects of P. guajava L. extract on human keratinocytes (HaCat). However, based on the results obtained in this study, it is possible to conclude that P. guajava L. extract, at different concentrations, does not induce micronucleus formation, meaning it does not cause genetic alterations in human keratinocytes.

This study has some limitations as it is an in vitro study and was performed in laboratory settings, therefore, further studies should be conducted in vivo and clinical settings to make sure of the efficacy of P. guajava L. hydroethanolic extract as endodontic irrigant, or even as intracanal medication. Lastly, the findings of the present study are promising and indicate the potential for advancing to future clinical studies. The extract could eventually be used as an adjunct to current decontamination methods in combating microorganisms present in the root canal system.

Conclusions

Psidium guajava L. hydroethanolic extract contained bioactive compounds such as rutin, quercetin, and kaempferol and exhibited notable antioxidant potential (IC50 = 10.39 µg/mL). Besides, it demonstrated significant antimicrobial and antibiofilm activity against E. faecalis and C. albicans, with the most effective concentration being 8.37 mg/mL. It showed lower cytotoxicity and genotoxicity than conventional irrigants, suggesting better biocompatibility. These findings highlight the potential of P. guajava L. extract as a natural antimicrobial agent in endodontics.

Supplemental Information

Supplemental Information 1 Cytotoxicity data.

Supplemental Information 2 Antibiofilm data.

Additional Information and Declarations

Competing Interests

Amjad Abu Hasna is an Academic Editor for PeerJ.

Author Contributions

Lara Steffany de Carvalho conceived and designed the experiments, performed the experiments, analyzed the data, prepared figures and/or tables, and approved the final draft.

Vanessa Marques Meccatti-Domiciano performed the experiments, prepared figures and/or tables, and approved the final draft.

Livia Ramos Dorta da Silva performed the experiments, prepared figures and/or tables, and approved the final draft.

Maria Cristina Marcucci performed the experiments, authored or reviewed drafts of the article, and approved the final draft.

Cláudio Antonio Talge Carvalho conceived and designed the experiments, authored or reviewed drafts of the article, and approved the final draft.

Amjad Abu Hasna conceived and designed the experiments, performed the experiments, analyzed the data, prepared figures and/or tables, authored or reviewed drafts of the article, and approved the final draft.

Luciane Dias de Oliveira conceived and designed the experiments, analyzed the data, prepared figures and/or tables, authored or reviewed drafts of the article, and approved the final draft.

Data Availability

The following information was supplied regarding data availability:

The raw measurements are available in the Supplemental Files.

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
