# Peer review of "Psidium guajava L. hydroethanolic extract as endodontic irrigant: phytochemical analysis, antioxidant activity, antimicrobial action and biocompatibility"

_PeerJ, doi:10.7717/peerj.19301_

## Round 0.1 · original submission · Major Revisions

· Academic Editor

Major Revisions

The reviewers have raised concerns regarding the clarity and completeness of the methodology, particularly in the extract preparation, analytical techniques (e.g., HPLC peak identification), and antimicrobial assays. Additionally, the discussion needs to provide a stronger comparison with previous studies, better highlight the novelty of the findings, and address study limitations. Language and formatting issues, including grammatical errors and inconsistencies, should also be corrected.

In addition to the reviewers' feedback, I would like to highlight the following points:

Abstract: There is no need to describe in detail the statistical analysis, the software used and the p threshold in the abstract.

The manuscript contains typographical errors that need correction (e.g., "irrgiants" in Line 101).

The rationale for selecting the HaCaT cell line for cytotoxicity testing should be clarified.

The methodology should explicitly describe how molecule identity was attributed to peaks in the chromatogram.

Line 299: What is the point of using multiple normality tests? The authors should choose one and use only it.

The conclusions section is very poorly written. It is also concerning that authors conclude the presence of specific molecule solely based on retention times from the literature. It is not clear what IC50 refers to in the conclusion.

The manuscript should mention the limitations of the study.

·

Basic reporting

Figure 1 quality is low.

Experimental design

Not all methods are described with sufficient detail and information to replicate.
Please see additional comments for details.

Validity of the findings

Novelty not well assessed.
Conclusion are not well stated.
Please see additional comments for detail information.

Additional comments

Lines 118-120: The null hypothesis needs to be restructured. Studies by Baldoni et al. (2023) and other related literature not mentioned by the authors have demonstrated the antimicrobial activity of the extract.

Baldoni et al. (2023) should be cited in the plant extract preparation section. In line 128, the author should specify the processing steps applied to the leaves before and after collection, prior to drying. Additionally, the type of filter paper used should be clearly mentioned. At the end of line 134, add: "and was stored for further analysis."

Total Flavonoid Content:
There is a contradiction in the methodology. In line 148, a stock solution of 100 µL was prepared in a 10 mL volumetric flask. However, in line 150, a 200 µL aliquot was taken from the stock. If the aliquot was prepared by combining two parts of the 100 µL stock solution, it should be clarified. Otherwise, the correct volume should be stated (e.g., "300 µL of the extract was prepared" or "two parts of the triplicate stock solution (200 µL) were used to prepare the aliquot"). Additionally, specify the instrument used to measure the absorbance of the sample.

Total Phenol Content:

In line 161, replace "small amount" with a quantified value.
In line 164, remove the word "approximately."
In line 168, specify which type of water was used to complete the volume.
In line 169, provide details on how "the final volume was adjusted."
In line 170, specify the type of spectrophotometer used (e.g., UV-Vis, AAS, etc.).
HPLC Analysis:
In line 176, describe how the sample was prepared or reconstituted for analysis. Specify which sample was used and the preparation method.

Free Radical Suppressing Activity:

In lines 189-190, clarify what was obtained from the soluble solid content (expressed in g or mg) and explain how it was used to prepare the V/V %.
In line 193, specify the volume of ethanol and extract added, in sequence.
In line 195, quantify the amount of DPPH added to each test tube (1 to 10).
Antimicrobial Assays:
Include a citation or guideline for Minimum Inhibitory Concentration (MIC) and Minimum Microbicidal Concentration (MMC) determination.

Antibiofilm Activity:

In lines 241-244, provide details on how the 2% CHX and 2.5% NaOCl solutions were prepared and cite relevant sources.
Express all concentrations in mg/mL instead of percentages throughout the manuscript and figures for proper comparison with sample concentrations.
Cytotoxicity Evaluation:
Include a citation or guideline for cytotoxicity evaluation.

Discussion Section:

Lines 371-378 should be removed. The null hypothesis should be mentioned in the discussion section but preferably placed in the conclusion.
Lines 379-402: The identified phytochemicals should be consolidated into a single paragraph. Discuss trends in the chromatogram, highlighting which phytochemical is most abundant and presenting percentages. Compare these findings with related literature on Psidium guajava, justifying any discrepancies. Additionally, the resolution of the figure should be improved.
Lines 403-417: The antioxidant discussion can remain unchanged.
From line 418 onward, the discussion should be rewritten to analyze trends in Figures 2, 3, and 4. The antimicrobial activity should be correlated with the phytochemicals identified in the HPLC analysis.
Below is my observation on Microbial Viability Trends (C. albicans ATCC 18804 in Figure 2) but no information like this is captured in the discussions and that's where the NOVELTY is:
As the extract concentration increases, microbial viability decreases. However, its effect is not as strong as CHX and NaOCl due to the lower concentration of the extract in mg/mL. A key question is whether doubling the extract concentration from 8.37 to 16.64 mg/mL (or matching the mg/mL concentration of CHX and NaOCl) would reduce microbial viability beyond 80%. If so, would this increase cytotoxicity or genotoxicity? A similar trend should be harmonized for bacteria, fungi, micronuclei, and cellular viability. Comparing these results with related literature will highlight the novelty of the study. The author can refer to the following article for how ciprofloxacin was compared with P. guajava extract: https://doi.org/10.37256/fce.4120232370.

Conclusion Section:
The conclusion should be rewritten based on the improved discussion. If the results support its use in endodontics, it should be stated as a potential solution to the challenges mentioned in lines 93-101. Consequently, the background, abstract, and results section should also be revised accordingly.

Keywords:
Add "endodontics" and "HPLC" to the keywords.

Reviewer 2 ·

Basic reporting

Please find the attachment

Experimental design

Please find the attachment

Validity of the findings

Please find the attachment

Additional comments

Please find the attachment

Annotated reviews are not available for download in order to protect the identity of reviewers who chose to remain anonymous.

Reviewer 3 ·

Basic reporting

English language and grammar need to be revised. need to break the long sentences for more clarity.
work is supported with sufficient related literature.

Experimental design

research problem falls in Aims and Scope of the journal.
units of phenol and flavonoid are missing in material and method sections. other changes are mentioned in Pdf file.

Validity of the findings

It is better to give concentration of active compound in guava extract instead of extract concentration. concentration of active compound in extract is changed in different regions or in leaves of different ages or different varieties and this extract could not be effective as effective in present study. Otherwise, the purpose of your study is not clear by reading conclusion only.

Annotated reviews are not available for download in order to protect the identity of reviewers who chose to remain anonymous.

---

## Round 0.2 · accepted · Accept

· Academic Editor

Accept

The authors have addressed all of the reviewers' comments. Thus, the manuscript is accepted in its present form.

·

Basic reporting

Satisfactory

Experimental design

Satisfactory

Validity of the findings

Satisfactory

Additional comments

Line 202 to 204: replace with, High-performance liquid chromatography (HPLC) was used to characterize the marker content and phytochemical profile of the extract prepared inutility from the guava leaves.
But they author can rephrase to show are continuous flow of information, the current way is in a form of report not explanatory.

Reviewer 2 ·

Basic reporting

No comments

Experimental design

No comments

Validity of the findings

No comments

Additional comments

I am pleased to inform you that I have reviewed the authors' revised manuscript and their responses to my comments. The authors have addressed all the concerns raised during the review process in a satisfactory manner. Their thorough revisions and detailed explanations have clarified all issues, and the manuscript now meets the scientific standards of PeerJ.

Reviewer 3 ·

Basic reporting

Appropriate

Experimental design

well written material and methods

Validity of the findings

Conclusion had been modified and now it is well stated.

Additional comments

The article should be accepted in present form